# Effectiveness of Volatiles Emitted by *Streptomyces abikoensis* TJGA-19 for Managing Litchi Downy Blight Disease

**DOI:** 10.3390/microorganisms12010184

**Published:** 2024-01-17

**Authors:** Mengyu Xing, Tao Sun, Tong Liu, Zide Jiang, Pinggen Xi

**Affiliations:** 1Key Laboratory of Green Prevention and Control of Tropical Diseases and Pests, Ministry of Education, School of Tropical Agriculture and Forestry, Hainan University, Haikou 570228, China; xingmengyu@hainanu.edu.cn (M.X.); 990251@hainanu.edu.cn (T.S.); 993604@hainanu.edu.cn (T.L.); 2Department of Plant Pathology/Guangdong Province Key Laboratory of Microbial Signals and Disease Control, South China Agricultural University, Guangzhou 510642, China; zdjiang@scau.edu.cn

**Keywords:** secondary metabolites, antifungal, fruit, SEM, TEM

## Abstract

Litchi is a fruit of significant commercial value; however, its quality and yield are hindered by downy blight disease caused by *Peronophythora litchii*. In this study, volatile organic compounds (VOCs) from *Streptomyces abikoensis* TJGA-19 were investigated for their antifungal effects and studied in vitro and in planta for the suppression of litchi downy blight disease in litchi leaves and fruits. The growth of *P. litchii* was inhibited by VOCs produced by TJGA-19 cultivated on autoclaved wheat seeds for durations of 10, 20, or 30 days. Volatiles from 20-day-old cultures were more active in inhibition effect against *P. litchii* than those from 10- or 30-day-old cultures. These volatiles inhibit the growth of mycelia, sporulation, and oospore production, without any significant effect on sporangia germination. Additionally, the VOCs were effective in suppressing disease severity in detached litchi leaf and fruit infection assays. With the increase in the weight of the wheat seed culture of *S.abikoensis* TJGA-19, the diameters of disease spots on leaves, as well as the incidence rate and disease indices on fruits, decreased significantly. Microscopic results from SEM and TEM investigations showed abnormal morphology of sporangia, mycelia, and sporangiophores, as well as organelle damage in *P. litchii* caused by VOCs of TJGA-19. Spectroscopic analysis revealed the identification of 22 VOCs produced by TJGA-19, among which the most dominant compound was 2-Methyliborneol. These findings indicated the significant role of TJGA-19 compounds in the control of litchi downy blight disease and in improving fruit quality.

## 1. Introduction

*Litchi chinensis* Sonn., widely referred to as litchi, is a fruit that thrives in tropical and subtropical environments and is grown in more than twenty countries globally [1]. Rich in various nutrients including vitamins, amino acids, flavonoids, and trace elements, litchi’s pulp provides numerous health advantages. This makes litchi a common option both as a fresh fruit and in traditional Chinese medicine, and its significance is emphasized in China. *Peronophythora litchii*, the causal agent of litchi downy blight, is a highly destructive disease that affects litchi trees during their growth and developmental stages. This disease is widely distributed in litchi-growing regions and leads to the decay of nascent leaves, blossoms, and fruits [2]. Moreover, litchi downy blight is the primary postharvest disease of litchi, resulting in significant damage to the fruit and a significant reduction in its shelf life. In fact, it has been reported to cause a loss of 20–30% of litchi fruit annually [3].

The management of litchi downy blight is predominantly dependent on the utilization of synthetic agrochemicals [4,5]. However, the use of potent synthetic chemicals during both onfield and post-harvest applications is progressively raising concerns regarding human health, environmental safety, and biodiversity. Consequently, there is a growing need to explore the potential of microorganisms or plant-derived bioactive secondary metabolites as environmentally friendly alternatives for the development of fungicides to control phytopathogenic *P. litchii* [6,7].

Biological control agents have garnered significant interest as a promising avenue for the exploration of biological pesticides, owing to their abundant production of bioactive secondary metabolites possessing antifungal, antibacterial, and nematocidal attributes [8,9]. Volatile organic compounds (VOCs) produced by microbes have particularly captured attention, as they have been demonstrated to effectively regulate certain plant diseases [10,11]. For instance, volatile organic compounds (VOCs) derived from *Bacillus* spp. and *Pseudomonas fluorescens* can reduce crop diseases [12,13]. Additionally, VOCs produced by *Trichoderma koningiopsis*-51 have exhibited strong inhibitory activity against *Botrytis cinerea* and *Fusarium oxysporum*, and VOCs derived from *Gluconobacter cerinus* and *Hanseniaspora osmophila* have shown control effects against table grape-rot pathogens [14,15].

Studies have shown that compounds emitted by *Streptomyces* significantly alter the conidiophore and hyphal structures of various fungi and inhibit conidia germination [16,17,18,19]. For example, volatile compounds produced by *Streptomyces platensis* F-1 were reported to have potent antifungal activities against several plant fungal diseases such as strawberry fruit rot, rice seedling blight, and oilseed rape leaf blight [20]. Li et al. [21] demonstrated the antimicrobial activities of VOCs emitted by *S*. *globisporus* JK-1 against *Botrytis cinerea* infection on tomatoes. Similarly, VOCs of *S*. *coelicolor* were effective against *Penicillium chrysogenum* and *B. cinerea*. These volatiles significantly inhibited the growth of mycelium and spore germination [22]. In another study, the membrane of *Fusarium moniliforme* mycelium was affected by VOCs produced by *S. philanthi* TD-1, which have the ability to disrupt the mycelial membrane permeability of *Fusarium F. moniliforme* [23]. Furthermore, several other biocontrol agents, including bacterial and fungal biocontrol agents, were reported to produce biologically active VOCs against a range of plant pathogens including *P. litchii* [24,25,26]. However, the potential of VOCs from *S. abikoensis* in managing downy blight on litchi remains yet to be explored.

In this research, we focused on investigating the in vitro efficacy of *S. abikoensis* TJGA-19 in combating *P. litchii*. Additionally, we identified the VOCs through solid-phase microextraction gas chromatography–mass spectrometry (SPME-GC–MS) analysis. The efficacy of these VOCs to minimize the losses in litchi during postharvest storage was investigated. The results of this study suggest that VOCs produced by *S. abikoensis* TJGA-19 hold promise for efficiently controlling post-harvest litchi downy blight and improving the fruit quality.

## 2. Materials and Methods

### 2.1. Microorganisms and Plant Materials

A total of 163 actinomycetes were isolated from Rhizosphere soils of wild litchi in the Bawangling Primeval Forest Reserve, located in Haikou City, Hainan Province, China [27]. Additionally, 17 plant pathogens (Appendix A) were obtained from the fungus laboratory at the Department of Plant Pathology, South China Agricultural University, with the courtesy of Professor Zide Jiang and maintained on Potato Dextrose Agar (PDA).

Leaves of litchi (cv. Huaizhi) were obtained from the litchi orchard at the South China Agricultural University in Guangzhou, China. The tender leaves were carefully chosen based on their consistent shape, size, and absence of any diseases. Litchi fruits (cv. Baitangying) at approximately 80% maturity and of uniform size, without any pests or diseases, were collected from a litchi planting orchard in Haikou City, Hainan, China. Both leaves and fruits were washed under running water and allowed to air-dry before inoculation.

### 2.2. Screening Antagonistic Actinomycetes for Volatile Mediated Antifungal Activity against P. litchii

A double-chamber device was used to investigate the volatile-mediated antifungal effects of actinomycete strains on PDA plates. The actinomycete strains were cultured through streaking on PDA plates (φ = 9 mm) and incubated at 25 °C for 7 days. Subsequently, the actinomycete-inoculated plates were covered (face down) with another PDA plate containing a *P. litchii* mycelial plug of 6 mm. To facilitate gas exchange and stop outside air contact, these plates were tightly sealed with parafilm and kept at 25 °C for six days. A control group was established with only *P. litchii* in the plates. The diameter of *P. litchii* growth was measured through the cross-section method and the inhibition rate was calculated as inhibition rate (%) = [(growth diameter of control − growth diameter of treatment culture)/(growth diameter of control − diameter of the inoculated fungal cake)] × 100. The investigation was repeated three times, with three replicates each.

### 2.3. Production of S. abikoensis TJGA-19 Volatile Substances

In order to produce volatile substances, the strain TJGA-19 was cultivated in PDB medium at 25 °C, with agitation at 180 rpm for a duration of 5–7 days. Subsequently, the concentration of TJGA-19 spore suspension was maintained at 1 × 10^7^ spores/mL by diluting it with PDB medium. The resulting spore suspension was poured onto the sterilized wheat seeds in 1000 mL conical flasks, at a ratio of 1 mL per 100 g of wheat seeds.

### 2.4. The Effect of Cultivation Time on the Antifungal Activity of Volatiles Produced by S. abikoensis TJGA-19

An antifungal bioassay was conducted using a specific methodology described by Li et al. [21] with minor modifications. Briefly, the experimental setup involved placing four small Petri dishes (60 mm diameter, 15 mm height) inside a larger Petri dish(150 mm diameter, 30 mm height, with a 0.5 L inner volume). Among the smaller dishes, three were filled with 5 mL of PDA medium that had been inoculated with a fungal plug (6 mm diameter) obtained from the periphery of *P. litchii*. The fourth dish contained wheat seed cultures of TJGA-19, with a concentration of 40 g/L, which had been inoculated for 10 d, 20 d, and 30 d. Sterilized wheat seeds 40 g/L without TJGA-19 inoculation were used as a control. The bigger plates were covered with a lid and tightly sealed with parafilm, permitting for unhindered gas exchange between the colonies within the dish while simultaneously avoiding direct contact, and the colony diameter on each plate was measured after incubation at 25 °C for 6 days. The percentage inhibition of fungal growth was determined using the above equation mentioned in Section 2.2. The experiment was conducted thrice.

### 2.5. Testing Antifungal Spectrum of Volatiles of S. abikoensisTJGA-19

The antifungal potential of volatiles from *S. abikoensis* TJGA-19 was tested in vitro through a method described previously in Section 2.2 with slight modification. TJGA-19 (60 g/L) culture, grown on wheat seeds, was placed in a Petri plate and exposed to another PDA plate (upside down) containing a pathogenic mycelial plug of 6 mm. The evaluation was conducted separately for 17 pathogens. Sterilized wheat seeds (60 g/L) without TJGA-19 inoculation were used as a control group. After incubating at 25 °C for six days, the colony diameter and inhibition rate was calculated as described above in Section 2.2. The investigation was repeated thrice with three replicates each.

### 2.6. Preparation of P. litchii Sporangia Suspension

To prepare *P. litchii* sporangia suspension, the fungus was cultivated on a carrot juice agar (CA) medium (juice extracted from 0.2 kg of carrots was diluted with water to a total volume of 1 L and subsequently supplemented with 15 g of agar) at a temperature of 25 °C for a duration of 6 days. Subsequently, the cultivated fungus was eluted with sterile water and filtered using four layers of sterile gauze.

### 2.7. Inhibitory Activity of Volatiles from S. abikoensisTJGA-19 In Vitro

In this assay, the inhibitory effects of volatiles from *S. abikoensis* TJGA-19 on various aspects of *P. litchii* were investigated, including mycelial growth, sporulation, sporangial germination, and oospores production. The effect of these volatiles on radial growth and sporangial production was assessed using the antifungal bioassay method, as described in Section 2.4. The weight of the wheat seed culture of TJGA-19 ranged from 2 g/L to 40 g/L, the diameter of the fungal colony was measured after incubating for six days at 25 °C, and the total number of sporangia per plate was determined. Non-TJGA-19-inoculated 16 g/L autoclaved wheat seeds were used as a control. Each treatment was replicated three times.

The effect of volatiles on the germination of *P. litchii* sporangia was investigated. Sporangia suspensions, consisting of 5 × 10^4^ sporangia/mL, were collected from 6-day-old cultures. Subsequently, 2 µL droplets of the sporangia suspension were placed on 3 mL of PDA in three smaller dishes, with each dish containing five points. These dishes were then subjected to treatment with volatiles according to the aforementioned procedure. After incubating at 25 °C for 3 h, the sporangia were observed using an optical microscope in order to assess the rate of germination. A sporangium was deemed to have germinated when the length of its germ tube reached 1.5 times the diameter of the sporangium’s shorter side. A total of 250 sporangia were tallied for each treatment. Three sets of larger plates were prepared for each treatment, with the experiment being repeated three times.

To assess the impact on oospore production, *P. litchii* was introduced to CA medium in Petri dishes with diameters of 60 mm. The dishes were then subjected to fumigation using volatile organic compounds generated from varying quantities of wheat seed culture of TJGA-19, ranging from 1 g to 40 g per liter of airspace in the treatment containers. Non-TJGA-19-inoculated wheat seeds were utilized as the control group. Following a 14-day incubation period in darkness, plugs measuring 1 × 1 cm were excised from a 10 mm radius around the inoculation site, and oospore quantities were enumerated using a microscope. Each treatment was replicated independently three times.

### 2.8. Observation of Morphology by Electron Microscopy

*P. litchii* was subjected to a 3-day culture period, followed by fumigation with a wheat seed culture of TJGA-19 (100 g/L) using a double chamber device for an additional 3 days. Subsequently, a fungus plug measuring 8 mm in length, 5 mm in width, and 3 mm in height was excised from the periphery of the colony, three fungus plugs were obtained from each treatment and promptly immersed in pre-cooled 4% glutaraldehyde fixing solution. The samples were then stored overnight at 4 °C for preparation of scanning electron microscopy. For transmission electron microscopy observation, the mycelium of *P. litchii* was gently scraped with a sterile toothpick and it was placed in pre-cooled 2.5% glutaraldehyde for fixation. An equal amount of sterile wheat seeds were used as a control. The experiment was repeated three times.

### 2.9. Inoculation Bioassay In Vivo

The inoculation bioassay was performed as described in a previous study with minor modification [25]. In brief, place a layer of filter paper at the base of a Petri dish with a diameter of 200 mm and a height of 30 mm. Spray 8 mL of sterile water onto the filter paper (φ = 180 mm) to provide moisture. Position Petri dish (φ = 90 mm) in the center of the filter paper, containing varying quantities of *S. abikoensis* TJGA-19 wheat seed culture (8–32 g/L) and keep wheat seeds without TJGA-19 inoculation as a control group. Subsequently, arrange 10 leaves along the outer edge of the filter paper, ensuring that the backside of the leaves faces upwards. Apply sporangia suspension (2 × 10^4^ sporangia/mL) at 2 μL to the vein and promptly seal the large culture dish. After 48 h of inoculation, measure the length of the lesion. Three replicates containing 30 leaves per replicate were used for each treatment. The fruits were inoculated with 10 µL of sporangial suspension and kept at 25 °C. Incidence rate and peel browning were evaluated at 72 h after inoculation. Three replicates containing 30 fruits per replicate were used for each treatment.

### 2.10. Collection and Analysis of Volatiles Produced by S. abikoensis TJGA-19

The volatiles emitted by *S. abikoensis* TJGA-19 after a 20-day incubation period were gathered and subjected to analysis using SPME-GC/MS. A 20 mL vial containing either 4 g of sterilized wheat seeds (as a blank control) or a wheat seed culture of TJGA-19 was sealed with aluminum foil and incubated at 25 °C for 12 h. Subsequently, the volatiles were collected by introducing a 100 μm PDMS SPME fiber (Supelco, Palo Alto, CA, USA) into the vial and allowing it to remain there for 30 min at 50 °C. The volatile compounds absorbed by the fiber were then desorbed for a period of 3 min at 250 °C and analyzed using an Agilent 7890B GC-MS (Agilent Technologies, Inc., Palo Alto, CA, USA) equipped with an HP-5MS column (30 m × 0.25 mm with a film thickness of 0.25 μm). The mass spectrometer was utilized in the positive electron ionization mode at an energy level of 70 electron volts and a temperature of 230 degrees Celsius. To verify the identity of the compound, the mass spectra and retention times were compared with those of established standards in the National Institute of Standards and Technology (NIST) Library. Furthermore, the volatile compounds originating from autoclaved wheat seeds were eliminated from the baseline. The experiments were repeated thrice.

### 2.11. Statistical Analysis

The data were subjected to analysis using SPSS 23 statistical software (SPSS, Chicago, IL, USA) in order to perform an analysis of variance (ANOVA). To compare the different treatments, Duncan’s multiple range test was employed at a confidence level of 95% (*p* ≤ 0.05).

## 3. Results

### 3.1. Screening Actinomycetes for Antagonistic Activity

By using the double-chamber device method, 4 strains of actinomycetes were screened out from 163 strains, namely TJGA-19, TJSCA-71, TJSCA-17, and TJSIM-13, which significantly inhibited the growth of *P. litchii*. Among them, TJGA-19 exhibited the strongest antifungal effect, with an inhibition rate of 86.3% (Figure 1). Therefore, TJGA-19 was selected for further experiments. The strain TJGA-19 was identified as *S. abikoensis* in our previous study [27].

### 3.2. The Antifungal Activity of Volatile Substances from S. abikoensis TJGA-19 at Different Cultivation Times against P. litchii

The VOCs emitted by 10-, 20-, and 30-days-old wheat seed cultures of TJGA-19 were tested for their antifungal effect against *P. litchii*. The findings of this study indicate that *S*. *abikoensis* TJGA-19, when cultured on wheat seed for a duration of 20 days, exhibited the highest level of inhibitory effect (74.1%). In contrast, the antifungal rates observed for the 10-day and 30-day cultures were 54.9% and 63.7%, respectively, which were significantly lower than that from the 20-day culture (Figure 2). Hence, the *S. abikoensis* TJGA-19 wheat seed culture cultured for 20 days was used for further research.

### 3.3. Determination of the Antifungal Spectrum of S. abikoensis TJGA-19 Volatiles

To understand the antifungal spectrum of *S. abikoensis* TJGA-19 volatiles, double-chamber devices were used. The results showed that the volatile substances produced by TJGA-19 can inhibit the growth of 14 plant pathogens, among which the strongest inhibitory effect was on *P. litchii*, *Pythium myriotylum*, *Alternaria alternata*, *Colletotrichum musarum*. In these cases, the mycelium does not grow at all, with an inhibition rate of 100%. Next are *Phytophthora capsici* and *P. colocasiae*, with inhibition rates of 93.3% and 82.4%, respectively. However, the volatiles had almost no inhibitory effect on *F. oxysporum* f. sp. *cubense* Race 4, *F. oxysporum* f. sp. *cucumerinum* and *Botryosphaeria berengeriana* (Appendix A).

### 3.4. Antifungal Activity of Volatiles from S. abikoensis TJGA-19 on P. litchii

Volatiles emitted from the wheat seed culture of TJGA-19 caused significant suppression of sporulation and mycelial growth of *P. litchii*, whereas sterilized wheat seed did not affect the fungal growth. After 6 days of exposure to the wheat seed culture of TJGA-19 at 2, 4, and 8 g/L, the diameters of the fungal colonies were 46.8 mm, 46.2 mm, and 45.4 mm, respectively, and the sporulation counts were 124.7 × 10^4^, 123.5 × 10^4^, and 78 × 10^4^ sporangia per plate, respectively. On the other hand, the mycelial growth was 47.3 mm and the sporulation count was 129.2 × 10^4^ sporangia per plate in the controls. *P. litchii* was exposed to 12, 16, 24, and 32 g/L, and 40 g/L wheat seed culture of *S. abikoensis* TJGA-19; the colony diameters were 39.6 mm, 33.2 mm, 26.3 mm, 20.6 mm, and 1.22 mm, respectively. The sporulation counts were 51.3 × 10^4^, 5.3 × 10^4^, and 2.6 × 10^4^ per plate, and no sporangia were observed when the TJGA-19 wheat seed culture was applied at concentrations exceeding 32 g/L (Figure 3A–K). However, the fumigation treatment did not have any impact on sporangial germination (Figure 3L).

Subsequently, the oospore production under the influence of VOCs was investigated. Following a 14-day incubation period at 25 °C, the TJGA-19 cultures exhibited a dose-dependent suppression of *P. litchii* oospore production, with no oospores observed when treated with TJGA-19 culture concentrations exceeding 12 g/L (Figure 4A,B). In general, the outcomes of our study revealed that the volatiles emitted by TJGA-19 had a suppressive effect on the mycelial growth, sporulation, and oospore production of *P. litchii.*

### 3.5. Volatiles Resulted in Morphological and Ultrastructural Changes in P. litchii

The scanning electron microscopy results revealed that in the control treatment, the sporangia of *P. litchii* displayed a spherical shape, fullness, and a smooth cell wall. Additionally, mycelia and sporangiophores exhibited uniformity, slenderness, and a smooth and intact surface (Figure 5A–C). Conversely, when the mycelium of *P. litchii* was exposed to the volatile gas emitted by a 100 g/L *S. abikoensis* TJGA-19 wheat seed culture, the sporangia exhibited a rough and sunken cell wall, and the mycelia and sporangiophores experienced collapse and withering (Figure 5D–F).

The transmission electron microscopy findings demonstrated that the cell wall and the cytoplasm of the sporangia in the control group exhibited uniformity, with neatly arranged organelles and normal morphology of the nucleus (Figure 6A,B). Conversely, in the treatment involving a 100 g/L wheat seed culture, the mitochondria and nuclei were no longer observable, and the cell wall was thickened, while vacuoles experienced an increase in both quantity and size (Figure 6C,D).

### 3.6. The Inhibitory Effects of S. abikoensis TJGA-19 Volatiles on Litchi Downy Blight in Litchi Leaves and Fruits

Litchi downy blight is a common occurrence during various stages of litchi tree growth and development, particularly during the vulnerable periods of tender leaf, flower, and mature stages. This downy blight results in leaf and flower shedding, as well as fruit decay, leading to substantial economic losses. Consequently, we conducted experiments to assess the efficacy of VOCs in controlling litchi downy blight, utilizing litchi leaves and fruits as test subjects. The results of inoculation on litchi leaves demonstrated that the volatile compounds emitted by *S. abikoensis* TJGA-19 exhibited significant efficacy in managing litchi downy blight. After 48 h of inoculation, the mean lesion length reached 26.7 mm in control group. However, when the concentration of the wheat seed culture of *S. abikoensis* TJGA-19 was elevated from 8 g/L to 32 g/L, the lesion length decreased substantially from 21.9 mm to 3 mm (Figure 7).

The findings of this study indicate that the application of TJGA-19 volatiles through fumigation significantly mitigated disease severity on litchi fruits, particularly when higher concentrations of volatiles were employed (Figure 8B–E), in comparison to the control group (Figure 8A). The application of TJGA-19 sweat seed culture (32 g/L) resulted in a significant reduction in the incidence of litchi downy blight compared to the control (Figure 8F). Furthermore, the TJGA-19 VOCs demonstrated remarkable efficacy in delaying browning, as evidenced by significantly lower browning index values in the groups treated with TJGA-19 sweat seed culture (8, 16, 24, 32 g/L) compared to the control group (Figure 8G). These findings suggest that the use of TJGA-19 volatiles effectively inhibits the development of downy blight in both litchi leaves and fruits.

### 3.7. Analysis of Volatiles Produced by S. abikoensis TJGA-19

A total of 22 compounds were obtained by employing GC-MS detection to examine the volatile substances generated by *S. abikoensis* TJGA-19 over a period of 20 days. All of these compounds exhibited high similarity indices (SIs) exceeding 90% when compared to those present in the NIST library (Figure 9, Table 1). The majority of these compounds can be classified into various categories, including olefins, esters, organic acids, and alkanes. Notably, the compound with the highest detected concentration in this particular experiment was 2-Methyliborneol, followed by Dodecane, 2, 6, 10-trimethyl-, Tetradecane, 2, 6, 10-trimethyltridecane, and so forth.

## 4. Discussion

Microbes are increasingly familiar as promising and effective methods for controlling plant diseases in agricultural crops. They act as biocontrol agents, targeting various plant pathogens including fungi, oomycetes, nematodes, and bacteria. The role of VOCs produced by these microorganisms is of particular interest for their efficacy in managing plant diseases [28,29]. Among biocontrol agents, yeasts were reported to produce several key VOCs, including alcohols like ethyl alcohol, phenylethyl alcohol, and 3-methyl-1-butanol and esters such as ethyl acetate and isoamylacetate, which are effective in controlling several fungal diseases on plants [11]. Likewise, numerous strains of *Streptomyces* spp. are reported to emit a broad spectrum of VOCs, ranging from alcohols to sesquiterpenes, esters, terpenes, aldehydes, and compounds containing sulfur and nitrogen. These VOCs are known not only for their antimicrobial activities against phytopathogens but also for their potential to enhance plant growth [30].

Various species of *Streptomyces*, such as *S. philanthi* RM-1-138 [31], *S. setonii* WY228 [32], *S. salmonis* PSRDC-09 [33], *S. fimicarius* BWL-H1, *S. globisporus* JK-1 [21,26,34], *S. coelicolor* [22], *S. philanthi* RM-1-138 [35], and *S. alboflavus* TD-1 [23], have been identified for their antifungal activities against a range of pathogenic fungi. These antimicrobial activities were attributed to their inhibition effects on mycelial growth, sporulation, and spore germination, either in vitro or through the suppression of disease progression in plants. In our research, we explored the antifungal effects of VOCs from *S. abikoensis* TJGA-19 on *P. litchii*. We found that VOCs from TJGA-19 caused significant inhibition of *P. litchii* growth in vitro and a notable decrease in the severity of downy blight on litchi fruits and leaves. *P. litchii* was exposed to 32 g/L and 40 g/L wheat seed culture of *S. abikoensis* TJGA-19; the colony diameters were 20.6 mm and 1.22 mm, which were significantly lower than those in the control groups. Notably, when the TJGA-19 wheat seed culture was administered at concentrations higher than 32 g/L, no sporangia were detected, but there were 129.2 × 10^4^ sporangia per plate in the controls. The oospores of *P. litchii* are significantly inhibited by the VOCs of TJGA-19. Upon exposure to TJGA-19 culture concentrations of 12 g/L, the production of oospores is nearly eradicated. These observations indicate that the VOCs of TJGA-19 exert a notable influence on the vegetative development, cellular differentiation, and pathogenesis of *P. litchii.* These effects are similar to those previously reported for VOCs emitted by *S. salmonis* PSRDC-09 against *Colletotrichum gloeosporioides* in chili fruits [33]. However, unlike other studies investigating the antifungal effects of *Streptomyces* spp., VOCs emitted by TJGA-19 did not significantly affect the sporangia germination in *P. litchii*. This could be due to the quick germination of sporangia, happening within 2 h, a timeframe too short for the concentration of VOCs to be effective. Importantly, this study firstly documents the antifungal activities of VOCs emitted by *S. abikoensis* against the phytopathogen *P. litchii*.

To understand how VOCs emitted by TJGA-19 impede the growth and development of *P. litchii*, a detailed scanning electron microscopy (SEM) study was conducted on *P. litchii* following exposure to VOCs. Results from SEM analysis showed that VOCs caused the sporangial surface of *P. litchii* to wrinkle and roughen, and led to deformations and concave collapses in the mycelia. Transmission electron microscopy (TEM) further highlighted significant destruction to some cellular organelles and cell wall thickening. Similar action mechanisms, confirmed by SEM and TEM analysis, were previously reported with pterostilbene, zeamines, isoliquiritin, hypothemycin [36,37,38,39], or VOCs produced by *S. fimicarius* BWL-H1.

Through GC-MS analysis, a total of twenty-two VOCs emitted by TJGA-19 were identified, with 2-Methyliborneol emerging as the most prominent, followed by Dodecane, 2, 6, 10-trimethyl-, Tetradecane, and 2, 6, 10-trimethyltridecane. However, the existing knowledge on the inhibitory effects of these compounds against plant pathogens is still not fully understood. It is possible that a singular substance may exert a significant influence, or alternatively, multiple substances may synergistically interact, necessitating further investigation.

## 5. Conclusions

Our study reveals that the VOCs produced by TJGA-19 exhibit significant antifungal effects against *P. litchii*. Furthermore, the degree of inhibition of *P. litchii*, both in vitro and in planta, correlates with the amount of TJGA-19 cultivated in wheat seed media. The exposure of *P. litchii* to VOCs caused severe destruction at the morphological and cellular levels. No damage on the litchi pericarp was observed after fumigating by volatiles for 3 days. This research highlights important insights into the antifungal mechanism of VOCs produced by *Streptomyces* species, offering potential approaches for controlling the discoloration and spoilage of litchi fruits caused by *P. litchii* during their shipment and storage. The lethal effects of TJGA-19 volatiles on *P. litchii* and other fungi suggest that fumigation with volatiles of *Streptomyces* could have widespread applications in controlling postharvest diseases of fruits, grains, vegetables, and other crops.

## Figures and Tables

**Figure 1 microorganisms-12-00184-f001:**
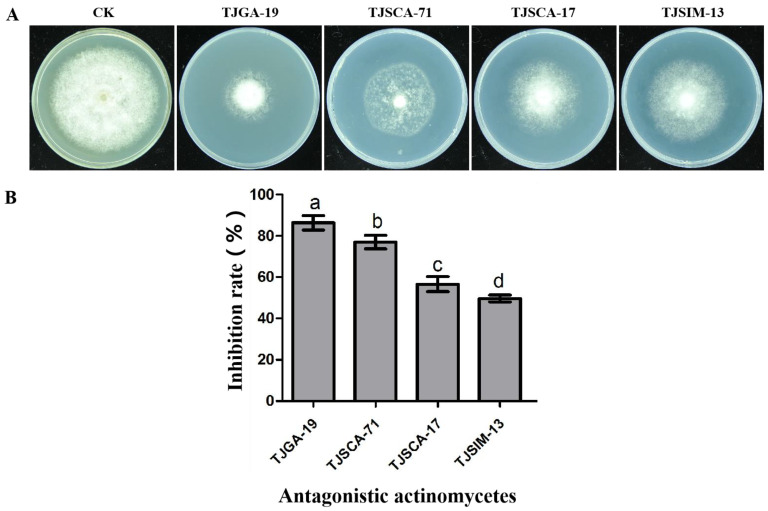
Screening antifungal actinomycetes against *P. litchii.* (**A**): The antagonistic effect of actinomycetes against *P. litchii*; (**B**): the inhibition rate was determined. The presented data represent the means ± standard error, and values followed by different letters were significantly different (*p* ≤ 0.05).

**Figure 2 microorganisms-12-00184-f002:**
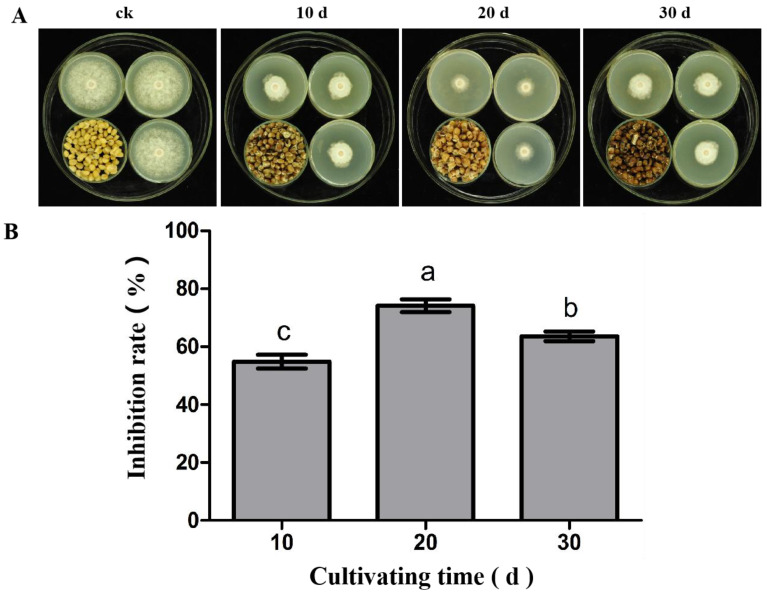
Effect of cultivating time on antifungal activity of volatiles produced by *S. abikoensis* TJGA-19. (**A**): The mycelial growth of *P. litchii* was observed after fumigation for 6 d using *S. abikoensis* TJGA-19, with different cultivation durations; (**B**): the determination of the inhibition rate. The data presented in this study represent the means ± standard error. The lowercase letters displayed above the bars in the accompanying graphs indicate significant differences, which were determined through statistical analysis using SPSS 23 with Duncan’s multiple range test (*p* ≤ 0.05).

**Figure 3 microorganisms-12-00184-f003:**
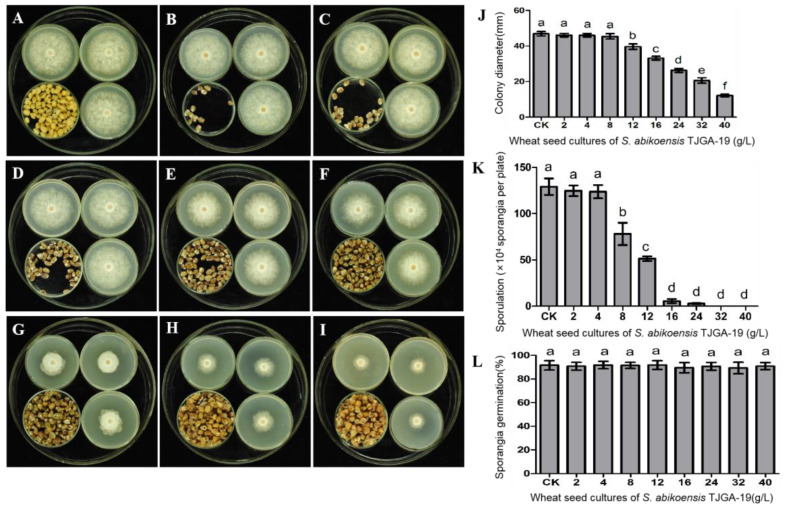
Effect of *S. abikoensis* TJGA-19 volatiles on mycelial growth, sporulation, and sporangial germination of *P. litchii.* (**A**–**I**): CK or *P. litchii* was subjected to fumigation with different concentrations (2, 4, 8, 12, 16, 24, 32, 40 g/L) of wheat seed culture of *S. abikoensis* TJGA-19 for a duration of 6 days and photographed; (**J**): colony diameter; (**K**): sporangial production; (**L**): sporangia germination rate. CK represents unfumigated control. Three independent experiments with triplicates were conducted for each instance, and values that were denoted by distinct letters exhibited significant differences (*p* ≤ 0.05).

**Figure 4 microorganisms-12-00184-f004:**
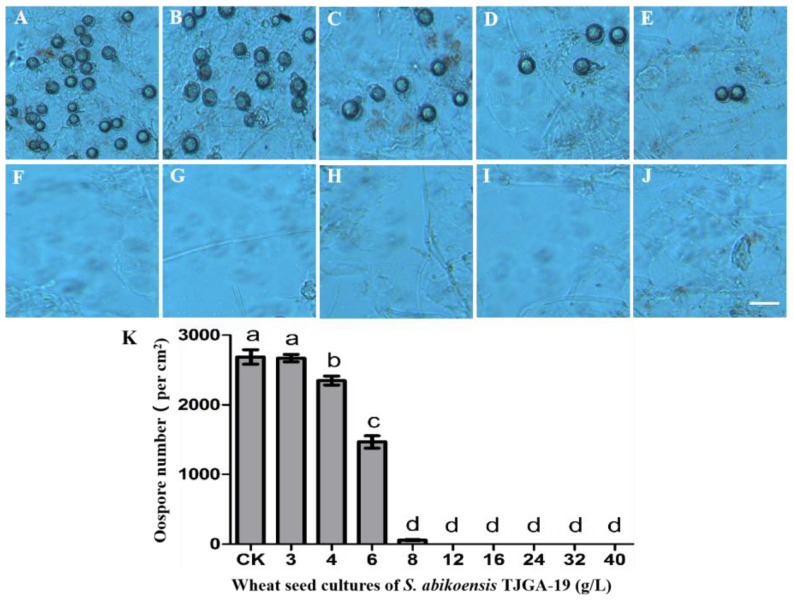
The effects of *S. abikoensis* TJGA-19 volatiles on oospore production. (**A**–**J**): The morphology of oospores was examined by capturing photographs subsequent to the fumigation process involving wheat seeds treated with either 16 g/L of wheat seed without the TJGA-19 culture or varying concentrations (3, 4, 6, 8, 12, 16, 24, 32, 40 g/L) of wheat seed culture of *S. abikoensis* TJGA-19 over a period of 14 days. (**K**): The oospore quantities were determined through calculation. The data presented herein are the result of three independent experiments, each conducted with triplicate samples. Statistical significance was determined by comparing values with different letters, with a significance level of *p* ≤ 0.05.

**Figure 5 microorganisms-12-00184-f005:**
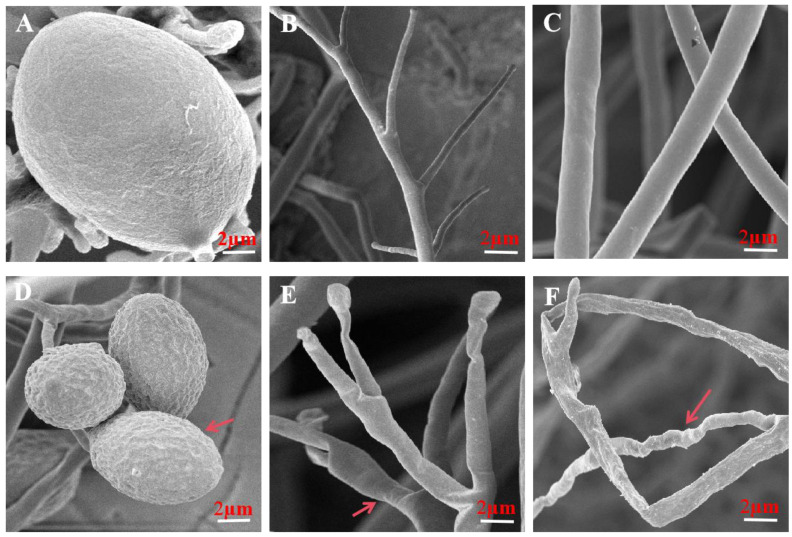
The volatiles of *S. abikoensis* TJGA-19 resulted in morphological alteration in the sporangia, hyphae, and sporangiophores of *P. litchii*. The untreated control group (**A**–**C**) exhibited normal morphology, whereas the group fumigated with a 100 g/L wheat seed culture of *S. abikoensis* TJGA-19 (**D**–**F**) displayed deformed sporangia, shrinking or distorted mycelia, and sporangiophores, as indicated by the arrows.

**Figure 6 microorganisms-12-00184-f006:**
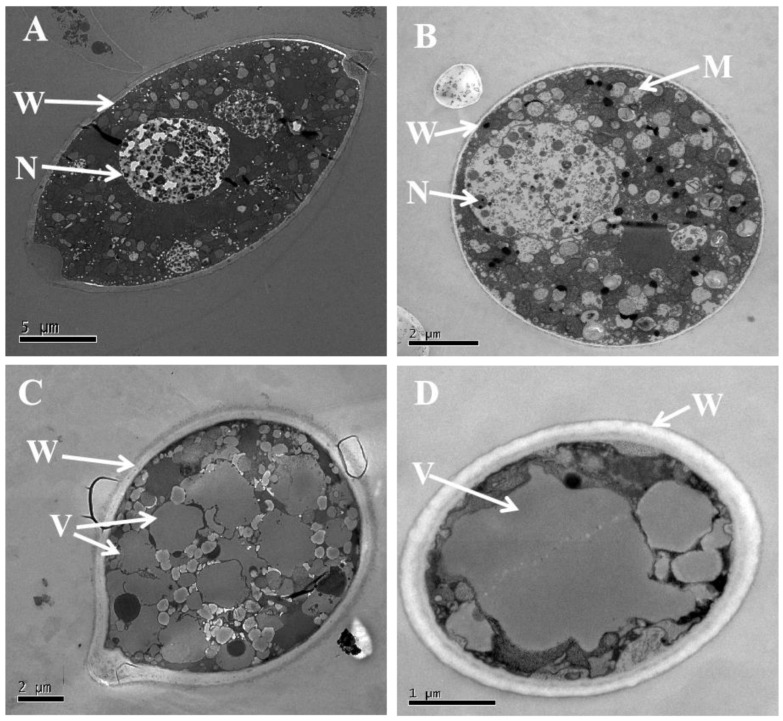
The cellular damage induced by the volatiles emitted by *S. abikoensis* TJGA-19. (**A**,**B**): Represented the untreated control group; (**C**,**D**): the mycelial samples that were subjected to fumigation with a 100 g/L wheat seed culture of *S. abikoensis* TJGA-19. Specifically, (**A**,**C**) corresponded to longitudinal sections through the *P. litchii* sporangium, whereas (**B**,**D**) represented tangential sections through the *P. litchii* mycelium. The abbreviations M, N, V, and W were used to indicate mitochondria, nuclei, vacuoles, and cell wall, respectively.

**Figure 7 microorganisms-12-00184-f007:**
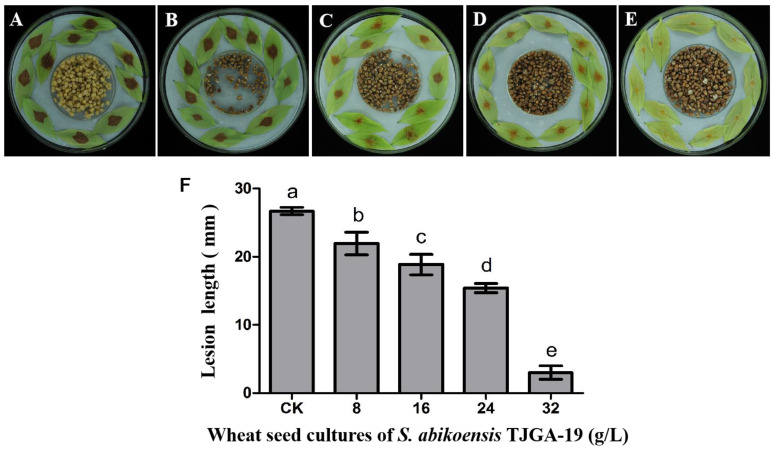
The antifungal efficacy of volatiles against *P. litchii* on detached leaves. (**A**): The control group was not subjected to fumigation; (**B**–**E**): leaves inoculated with *P. litchii* were fumigated with different concentrations (8, 16, 24, 32 g/L) of volatiles from the wheat seed culture of *S. abikoensis* TJGA-19; (**F**): lesion lengths were assessed after inoculation 48 h. The mean ± standard error (S.E.) was calculated based on three replicates. Values with different letters were found to be significantly different (*p* ≤ 0.05).

**Figure 8 microorganisms-12-00184-f008:**
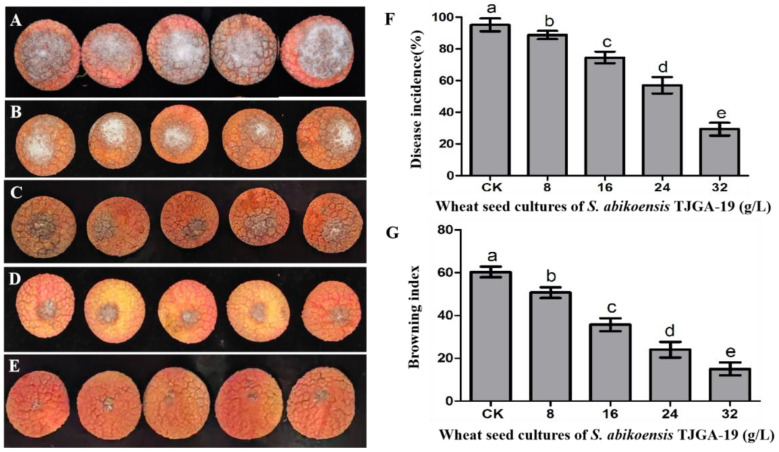
The antifungal properties of volatiles against the pathogen *P. litchii* on fruit. (**A**): Control; (**B**–**E**): fruits inoculated with *P. litchii* were fumigated with varying concentrations (8, 16, 24, 32 g/L) of volatiles from a wheat seed culture of *S. abikoensis* TJGA-19; (**F**): disease incidence; (**G**): browning index. Data presented are means ± standard error (S.E.) from three replicates per treatment. Values with different letters were found to be significantly different (*p* ≤ 0.05).

**Figure 9 microorganisms-12-00184-f009:**
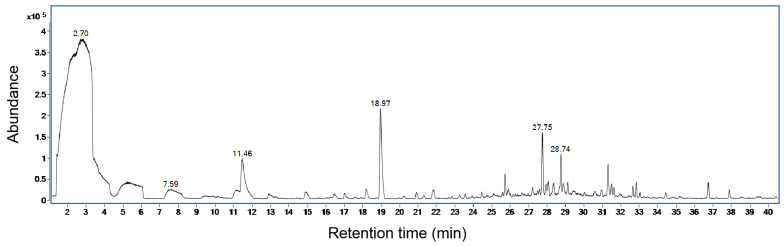
GC profile of *S. abikoensis* TJGA-19 volatile analysis.

**Table 1 microorganisms-12-00184-t001:** Volatile compounds from *S. abikoensis* TJGA-19 after 20 d incubation at 25 °C on autoclaved wheat seeds.

Volatile Organic Compounds ^a^	*RT* ^b^ (min)	Area (%)
Arsenous acid, tris(trimethylsilyl) ester	14.9147	3.5529
Benzonitrile, 2-(2-hydroxy-5-nitrobenzylidenamino)-	18.1902	3.2558
2-Methylisoborneol	18.9705	25.9287
4-(2-Methyl-cyclohex-1-enyl)-but-3-en-2-one	20.9163	1.5504
Cyclotetrasiloxane, octamethyl-	21.8342	2.8193
Tridecane	24.4551	1.6461
Naphthalene,1,2,4a,5,8,8a-hexahydro-4,7-dimethyl-1-(1-methylethyl)-, (1.alpha.,4a.beta.,8a.alpha.)-	25.7243	5.2184
1,3,5,7,9-Pentaethylcyclopentasiloxane	25.9002	2.1199
2-Methyltetracosane	27.2091	1.9879
Benzoic acid, hexyl ester	27.585	2.3067
Dodecane, 2,6,10-trimethyl-	27.7489	9.9877
4-(1-methyl-1-cyclobutyl)phenol	28.0618	4.9239
(2Z,4E)-3,7,11-Trimethyl-2,4,10-dodecatriene	28.3478	3.0363
Tetradecane	28.7473	7.8256
(1R,4aS,8aR)-1,4a-Dimethyl-7-(prop-1-en-2-yl)-1,2,3,4,4a,5,6,8a-octahydronaphthalene	28.8855	2.9132
Disparlure	29.1225	2.3421
2,6,10-Trimethyltridecane	31.3051	6.2293
1H-Cyclopropa[a]naphthalene, 1a,2,3,3a,4,5,6,7b-octahydro-1,1,3a,7-tetramethyl-, [1aR-(1a.alpha.,3a.alpha.,7b.alpha.)]-	31.4971	3.8272
Octacosane	32.6604	2.1236
Pentadecane	32.8364	2.234
Hexadecane	34.4358	1.2136
Longifolene	36.7445	2.9574

Notes: ^a^ Compounds detected in autoclaved wheat seeds or having less than 0.5% of total area are not included in the table. ^b^ RT, retention time.

## Data Availability

The authors of this article will provide the raw data that support the conclusions, without any unwarranted hesitation.

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
