# Peer review of "Effectiveness of Volatiles Emitted by Streptomyces abikoensis TJGA-19 for Managing Litchi Downy Blight Disease"

_microorganisms, 2024, doi:10.3390/microorganisms12010184_

Round 1

Reviewer 1 Report

Comments and Suggestions for Authors

I want to express my pleasure for being chosen to review the manuscript titled: “Effectiveness of volatiles emitted by Streptomyces abikoensis 2 TJGA-19 for managing litchi downy blight disease."

I have looked at the manuscript. Informative topic and scientific conclusion on the use of volatiles emitted by Streptomyces abikoensis 2 TJGA-19 for managing litchi downy blight disease   

I recommended major revision as the following

The manuscript should be revised for grammatical and style errors.

  • Abstract section: In the abstract, the research gap should be improved to strengthen the motivation of the work.

·         The novelty of the study needs to be highlighted compared to other similar studies.

·         Line 14 (planta) what ?

·         The authors mentioned that  (In this study, volatile organic compounds 12 (VOCs) from Streptomyces abikoensis TJGA-19 were investigated for their antifungal effects and stud- 13 ied in vitro and in planta for the suppression of litchi downy blight disease in litchi leaves and fruits)  However, there is no data showing experiments on the plant?  Must be clarified.

·         The authors mentioned that  (effective in suppressing diseases severity and prolonging the shelf life of litchi fruits) It must be clarified how the fruits were treated, what the evidence of shelf life of litchi fruits , and where the results of the experiments on the fruits ?

·         More details should be mentioned about the dates of sampling and in what part of the plant they were taken.

·         The conclusion is poorly; I think author should try to link better their work; I mean, the results should be quantitatively reported to present these potential applications better.

·         I suggest the authors to go through the manuscript one more time to minimize some errors, typos etc.

·         Please add all abbreviations mentioned in the manuscript to the abbreviation list at the end of the manuscript.

  • The discussion needs enhancement with real explanations not only agreements and disagreements. Authors should improve it by the demonstration of biochemical/physiological causes of obtained results. Instead of just justifying results, results should be interpreted, explained to appropriately elaborate inferences.
  • it is good to include some recommendations.
  • please check again the references and add the DOI if possible.

Author Response

Comment: I want to express my pleasure for being chosen to review the manuscript titled: “Effectiveness of volatiles emitted by Streptomyces abikoensis TJGA-19 for managing litchi downy blight disease."

I have looked at the manuscript. Informative topic and scientific conclusion on the use of volatiles emitted by Streptomyces abikoensis TJGA-19 for managing litchi downy blight disease.

I recommended major revision as the following

Response: Thank you for your time and providing useful suggestions for the improvement of our manuscript. We believe that these suggestions will increase the quality of presentation of our study. We have revised our manuscript according to your provided suggestions.

Comment: The manuscript should be revised for grammatical and style errors.

Response: The whole text has been revised for grammar and other language related mistakes.

Comment: Abstract section: In the abstract, the research gap should be improved to strengthen the motivation of the work.

Response: We have revised the abstract and improved the research gap as suggested

Comment: The novelty of the study needs to be highlighted compared to other similar studies.

Response: Done as suggested

Comment: Line 14 (planta) what ?

Response: This mean that the investigation was conducted both in vitro and in planta

Comment: The authors mentioned that  (In this study, volatile organic compounds 12 (VOCs) from Streptomyces abikoensis TJGA-19 were investigated for their antifungal effects and studied in vitro and in planta for the suppression of litchi downy blight disease in litchi leaves and fruits)  However, there is no data showing experiments on the plant?  Must be clarified.

Response: In planta mean the study was conducted in plants or their parts. Fruits and leaves are the parts of plants. There are several papers published in reputable journals in which this term is used for study in plant parts such as fruits or leaves.

Comment: The authors mentioned that (effective in suppressing diseases severity and prolonging the shelf life of litchi fruits) It must be clarified how the fruits were treated, what the evidence of shelf life of litchi fruits, and where the results of the experiments on the fruits?

Response: These results are presented in Figure 8

Comment: More details should be mentioned about the dates of sampling and in what part of the plant they were taken.

Response: The leaves and fruits were samplded as described in section 2.9.

Comment: The conclusion is poorly; I think author should try to link better their work; I mean, the results should be quantitatively reported to present these potential applications better.

Response: Conclusion section has been rewritten

Comment: I suggest the authors to go through the manuscript one more time to minimize some errors, typos etc.

Response: Done as suggested

Comment: Please add all abbreviations mentioned in the manuscript to the abbreviation list at the end of the manuscript.

Response: There are two abbreviations for two types of culture media in the manuscript. If need to add them to the end of the manuscript.

Comment: The discussion needs enhancement with real explanations not only agreements and disagreements. Authors should improve it by the demonstration of biochemical/physiological causes of obtained results. Instead of just justifying results, results should be interpreted, explained to appropriately elaborate inferences.

Response: Discussion section has been improved as suggested

Comment: it is good to include some recommendations.

Response: Some more recommendations have been added.

Comment: please check again the references and add the DOI if possible.

Response: Done as suggested

Reviewer 2 Report

Comments and Suggestions for Authors

The manuscript looks scientifically sound and timely. Indeed, volatile compounds with antimicrobial/antifungal properties represent a promising alternative to fungicides and antibiotics. The present paper devoted to the use of volatile substances produced by Streptomyces abikoensis to manage Peronophythora litchii. The design of the study looks logic and correct. The methods and results are described comprehensively and the experiments could be repeated. At the same time, there are some minor drawbacks ought to be corrected before the MS acceptance:

1. I believe the reference list should be expanded and some important works describing the use of VOCs to combat pathogens, such as Jepsen et al., 2022 (Curr Res Microb Sci); Pérez-Corral et al., 2020 (Emir J Food Agric); Cuervo et al., 2023 (Microorganisms); Campos-Avelar et al., 2021 (Toxins) can be added. 

2. There are some problems with design of the MS text and the reference list that should be improved according to the Microorganisms requirements. Also,

(i). 'in vitro' and 'in planta' should be written in italic (line 14 and below)

(ii). lines 65-66: 'moniliforme', not 'moniliformei'

(iii). line 326: 'S. abikoensis' - italic (as well as line 392)

(iv). line 340: 'thecontrol'

(v). Also, there are some missing or excess punctuation marks (commas, dots).  

Comments on the Quality of English Language

English looks correct and is easy to understand. 

Author Response

Comment: The manuscript looks scientifically sound and timely. Indeed, volatile compounds with antimicrobial/antifungal properties represent a promising alternative to fungicides and antibiotics. The present paper devoted to the use of volatile substances produced by Streptomyces abikoensis to manage Peronophythora litchii. The design of the study looks logic and correct. The methods and results are described comprehensively and the experiments could be repeated. At the same time, there are some minor drawbacks ought to be corrected before the MS acceptance:

Response: Thank you for your encouraging comments and providing useful suggestions for the improvement of our manuscript. We have revised our manuscript according to your provided suggestions.

Comment 1. I believe the reference list should be expanded and some important works describing the use of VOCs to combat pathogens, such as Jepsen et al., 2022 (Curr Res Microb Sci); Pérez-Corral et al., 2020 (Emir J Food Agric); Cuervo et al., 2023 (Microorganisms); Campos-Avelar et al., 2021 (Toxins) can be added. 

Response: Yes, these references will help to discuss the recent progress on the studied topic. We have added all these references in revised version. 

Comment 2. There are some problems with design of the MS text and the reference list that should be improved according to the Microorganisms requirements. Also,

Response: The revised version has been improved.

(i). 'in vitro' and 'in planta' should be written in italic (line 14 and below)

(ii). lines 65-66: 'moniliforme', not 'moniliformei'

(iii). line 326: 'S. abikoensis' - italic (as well as line 392)

(iv). line 340: 'thecontrol'

(v). Also, there are some missing or excess punctuation marks (commas, dots).  

Response: We fixed all these mistakes and double checked the whole text.

Comment: English looks correct and is easy to understand. 

Response: Thank you

Round 2

Reviewer 1 Report

Comments and Suggestions for Authors

I see that the authors answered all questions and the article can be published in this form